# Light in the tunnel or just a train; impact of supply chain finance solutions on financial service providers' financial performance by mitigating financial risk

Munibah Munir[ORCID]*[a], Nousheen Tariq Bhutta[ORCID][a]

Department of Management Sciences, Capital University of Science and Technology, Islamabad, Pakistan

[a] These authors contributed equally to this work.
* munibahmunirahmad@gmail.com

**Data Availability Statement:** By using the following URLs others can access the dataset •https://www.adb.org/what-we-do/trade-supply-chain-finance-program/participating-banks •https://

## Abstract

Supply chain finance is newly emerging concept and grab attend of financial sercive providers, buyers and suppliers. This study empirically examines the impact of supply chain finance solutions (SCFS), banks financial risk on financial service providers' financial performance using panel dataset of Asian Development Bank registered countries (Pakistan, China and Bangladesh) from 2012–2021. By breaking new ground, supply chain finance solution index is developed by combining several solutions to measure its impact on financial service provider financial performance. The results show a significant impact of supply chain finance solutions on financial performance of financial service providers. Furthermore, by offering SCF solutions a bank is able to reduce its financial risk for the external parties (e.g., investors, shareholders) This research encourages financial service providers (banks) to embrace the supply chain finance solution to enhance financial performance and allows them to evaluate their supply chain finance solutions investments as a technique to mitigate financial risk.

## Introduction

The Supply chain finance is a newly inovative and emerging practice that has grabbed the attention of suppliers, buyers, and financial service providers [1–4]. Supply chain finance ensures efficient financial flow through the information and goods flow phases. Different research shows that it is a "win-win" situation for all actors (i.e., buyers, suppliers, and financial service providers) [5, 6]. In any economy, the major contributors to the growth and survival of the banking sector are financial service providers and managing buyer-supplier relationships to reduce risk. Financial service providers, specifically banks, focus on to enhance their involvement in financial side of the global supply chain by offering different supply chain finance solutions according to the need of the supply chain participants [7].

Nevertheless, a financial service provider faces many risks that can destabilize their financial position, jeopardizing the stability of the whole economy and the financial performance of the financial service provider [8].

www.sbp.org.pk/reports/annual/FSAFS/2023/2023.
htm •https://www.bankasia-bd.com/about/
annualreport •https://www.thecitybank.com/report/
annualreports •https://www.ebl.com.bd/annual-
reports •https://www.dutchbanglabank.com/
investor-relations/financial-statements.html
•https://www.eximbankbd.com/report/Annual_
Reports •https://www.mutualtrustbank.com/
regulatory-disclosure/financial-statements/ •https://
www.primebank.com.bd/index.php/home/
financial_reports •https://www.pubalibangla.com/
Annual-Reports.asp •https://www.southeastbank.
com.bd/?page=annual_reports •https://ir.
ctbcholding.com/html/financial_reports.php
•https://www.hsbc.com/investors/results-and-
announcements/annual-report.

**Funding:** The authors received no specific funding
for this work

**Competing interests:** The authors have declared
that no competing interests exist

Due to financial crisis, banking sector distress and mainly financial risk and performance of the banks. Financial risk may not be avoided to enhance good financial performance; these two interdependent components must be evaluated simultaneously [9].

Supply chain finance program main pillar are finance providers [3, 4, 10, 11]. All participants risk level in supply chain finance lower by finance providers, if any participant not payback loan to provider. That's why finance providers are risk-takers and generate profits by performing this activity [7]. Different studies show that initiatives of supply chain finance allow finance providers to have low-risk exposure by offering different but attractive discount rates [7, 12]. Financial service providers' financial risk may be influenced by supply chain initiatives [13].

However, it is still unclear how the supply chain finance solutions index influences financial service providers' financial risk. Some researchers suggested that financial services providers are evaluated by stakeholders based on their financial risk mitigation capability using supply chain finance solutions [14]. However, there is very limited empirical evidence regarding the supply chain finance impact on their banks financial risk. Since Financial risk is one of the widely used market-based measurement ways and grabs the attention of many company investors and shareholders [15, 16]. There is a need to figure out this research gap in the literature and whether by offering supply chain solutions, finance providers reduce/mitigate financial risk.

Underpinned by a risk management perspective. Morck [17], suggested that an organization's internal control system with risk propensity leads toward risk level. Financial service providers take the risk contingent upon financial performance with the condition of financial risk propensity. Risk sharing reduces the quality risk and enhances performance levels [17]. Finance providers offer a SCFS contingent upon financial risk as it enhances performance. This study mainly focuses on the supply chain finance solution impact on the firm performance of financial service providers (banks) with the moderating role of financial risk. Different research has highlighted this research as having a high impact on the successful use of supply chain finance solutions [13]. For example, according to Selvaraj, [13], initiative of supply chain financial play a critical role in the financial risk of finance providers. Financial service providers' financial risk reduces by initiating supply chain finance, but the impact of financial service providers' financial performance remains empirically unexamined. Furthermore, other than examination of direct impact of combined supply chain finance solutions on finance provider's performance, this study also analyzes the moderating role of financial risk of financial service providers. Thus, this study attempts to answer the following question:

1. What is the impact of supply chain finance solutions on financial service provider's financial performance?

2. What is the moderating role of financial risk on SCFS and finance providers' financial performance?

To answer these questions, collect and use secondary panel data from multiple sources to open a new window with significant insights into the abovementioned relationships. The scarcity of empirical studies and the importance of supply chain finance solutions in the survival and economic situation make this paper an appealing substance of research. First, Actors of supply chain finance has grabbed the attention [1, 17]; there is little research contribution that connects supply chain finance solutions impact with financial service providers finance performance. This research is the first to study the combined supply chain solutions relationship with financial service providers' financial performance. By identifying the combined supply chain finance solutions relationship with financial service provider financial performance, this

study gives future direction to scholars to further investigate the implications of financial performance on supply chain finance service users from buyers and suppliers as supply chain finance participants. Moreover, this study increases the supply chain finance solutions knowledge by investigating the impact of financial risk on financial service providers' financial performance. Therefore, the findings allow finance providers to evaluate their offered supply chain finance solutions based on different firm resources.

## Theoretical background

Bargaining power theory in which all parties are in an argumentative situation like contract writing, making agreements, or barraging contracts where one party has more influence over the other party. Bargaining power is the relative ability of parties to exert influence over each other in a situation [18]. In supply chain finance, bargaining is called bargaining if one party has strong power and exercises power to gain more over the weaker party [19]. Financial service providers may not determine the risk level by offering supply chain finance solutions, but how financial risk affects their financial performance leads to a strong bargaining power exercise.

With more financial visibility for the financial service provider, they will gain more from the transaction over financial risk [20]. When banks offer supply chain finance solutions and act as risk-taker for the other two parties (buyers and suppliers) they can influence the other parties when they make the contract or negotiate the agreement. Banks with strong bargaining power towards other parties suggested that as risk-taker, they are more in the situation to extract influence over others to get offset against risk [21]. Bargaining power impact on financial service provider performance will be clearer via financial risk level, which forces the other parties to do what they will otherwise be less willing to do, for example, more growth [22].

In this case, if all parties have equal bargaining rights, they perform perfect rights equally. In actual business or bank loan situation is different, and one party may have more power over the other. Banks hold longer cash and create a deficit for the other two parties that have to find finance solutions for different banks. In this case, banks have more bargaining power, and the other two parties have less access to credit, ultimately leading them to high costs [23]. The weak bargaining power of parties has forced them to bear more risk and cost through the use of finance [2]. In the case of financial service providers, banks put pressure on buyers and suppliers to accept late payments if they believe that other parties have resource slack. With this behavior, banks might be possible to increase their bargaining power as the other two parties prepare to formally file for bankruptcy [21]. As a financial service provider with strong bargaining power has an opportunity to have more favorable finance provider terms against other parties, a financial service provider may choose not to use power against a weak party due to concern about the party disruptions and overall will affect financial performance and growth [24].

## Literature review and hypothesis development

Supply chain management is collaborating and coordinating different supply chain parties' goods flow, information, and finance flow optimization [4]. Different studies contribute to it as supply chain finance (SCF) can create a win-win situation for all the participants like suppliers, buyers, and financial service providers [25, 26]. Supply chain finance has a significant intersection of trade finance and the supply chain management field. Overall, all supply chain movement is to convert the material and information flow into the desired form along with the effective use of financial flows for all the supply chain parties [10]. The basic idea of supply chain finance all partices of supply chain get mutal value [13]. Morck [17], added classic firm-

oriented practices in supply chain finance now extended to deal with cash conversion cycle, Weighted average cost of capital (WACC), cash flow management, receivables and payables.

When measuring a firm financial performance and different financial decision phenomena, the Tobin's Q ratio is one of the indicators used in literature. Wernerfelt [27], used Tobin's Q ratio to explain cross-sectional returns implying a proxy for risk. Mansyur [28], used Tobin's q to measure firm performance with the relative importance to measure share effect. In literature, the Tobins q ratio is not limited and is used to measure firm performance. All financial performance combines all the management factors that may be used for optimal profit achievement in any firm resources [29].

Bank's financial performance measurement, also known as profitability [30, 31], measured financial performance by using return on assets (ROA) and net interest margin (NIM). Kassi [32], examined USA banks' failure factors and concluded return on assets has a significant performance factor for banks' failure.

**H1**: Supply chain finance solutions have a positive impact on financial service providers' financial performance.

Common risks are liquidity, credit, market, and non-financial risks [33]. Out of all financial risks directly affect the company's financial position internally and in the market. Any fluctuation in return will lead to financial risk [34]. Due to financial movement in financial markets arises financial risk [35]. Usually, it is linked with leverage and risk about all the due obligations and liabilities or not meeting with current assets. Due to the asymmetry of information among service providers, financial risk increases, and supply chain finance reduce this fact and reduce the uncertainty effect [36, 37]. Many service providers may fail to evaluate small businesses and their financial perspectives with conventional financing solutions [38]. Conversely, supply chain finance is offered by service providers and many substantial relevant business solutions. Supply chain participants maintained transaction history and credit informaiton [39].

Financial risk includes credit, liquidity, and operating risks, contributing to financial performance volatility [40].

Bolton [41], examined that financial risk significantly impacts financial performance. Financial success is measured using ROA and ROE, net equity to total assets, gross debt to total assets, and loan-deposit ratio for financial risk management. For this analysis, the population was 10 Botswana commercial banks with secondary data set time 2011–2018 using different research tests (descriptive statistics, correlation, and regression analysis. Finding of the study was interest rate had a significant negative effect on ROA and ROE.

Malkiel [42], examined the financial risk relationship with ownership structure using Tobin's q ratio. They concluded that financial leverage and business risk significantly negative impact inside ownership structure and firm financial performance.

**H2**: Supply chain finance solutions enable the finance providers to reduce/mitigate their financial risk by improving their performance.

## Methodology

In this research, Asian Development bank registered countries, Pakistan, China and Bangladesh 25 banks data used as sample from 2012–2021. This research sample covers conventional banks and excludes Islamic and saving investment banks due to their different business models. It focuses on the country level rather than the individual level of firm and soltuions. All variables' data are collected on an annual base from the bank's official websites and annual reports of banks.

## Variables description

**Financial performance.** Financial performance is measured in two ways; internal and external. Internal financial performance is estimated using the ROA of the year. First, ROA is computed as the net income ratio to total assets [43]. The value has been transformed into a log of ROA to improve the data normality. The ROA is selected because the return on assets shows the management's ability to make a profit from the bank's assets

$$ROA = \frac{PAT}{TA} \times 100$$

ROA is the return on assets, PAT is profit after tax, and TA is total assets.

Second, the external market is measured by Tobin's Q. this ratio is consistent with the [44], established efficient market hypothesis. This ratio is used to measure the company's potential future growth and existing assets. Tobin's Q ratio measures the investor's future expectations with the business's current strategies for evaluation [43, 45–47].

$$Q\ ratio = \frac{Total\ market\ value\ of\ firm}{Total\ Assets\ Value}$$

Where the Q ratio is Tobin's Q, the company's total market value is outstanding stock and debt, and the value of the total assets is the replacement cost of the company's assets (book value) (Christensen et al., 2010).

**Supply chain finance index.** In literature, there is list of 21 supply chain finance solutions offered by banks (S1 Annex). Each bank website check and mark how many supply chain finance solutions offered by particular bank out of list of the supply chain finance solutions, if a bank offers any one solution like account receivables finance or an early payment discount program it is coded as 1. On the other side if bank is not offering any particular supply chain finance solution it is coded as 0. The Principal Component Analysis technique of [48], is applied to quantify the supply chain finance solutions index using supply chain finance solutions as proxies.

$$SCFSI_t = \beta_0 + \beta_1 SS_t + \varepsilon_t \tag{1}$$

Where $SCFSI_t$ represents the supply chain finance solutions index, $SS_t$ represents supply chain finance solutions.

**Financial risk.** Financial risk is the proability of losing profit based on the bank's financial characteristics [11]. Financial risk is the average of two risks, credit risk and liquidity risk, contributing to financial performance [40]. Credit risk is measured as capital adequacy ratio (CAR) and non-performing loans (NPL). At the same time, the liquidity ratio is measured as total loans divided by total deposits.

$$FR = \frac{CR + LR}{2}$$

Where FR is financial risk, CR is credit risk, and LR is liquidity risk.

*Control variables*. The research model includes bank-specific control variables. The bank-specific variables are tier 1 capital ratio (T1), loan-to-asset ratio (LA), other earnings assets (OEA), and bank size (BS). T1 is used to control for differences in banking sector development, while the other variables (LA, OEA, and BS) are used to control for systemic, idiosyncratic, and market risks of the bank.

## Empirical model

In this research, a panel data analysis is used. To measure the model variables:

$$FP_{it} = \beta_o + \beta_1 SCFSI_{it} + \beta_2 FR_{it} + \beta_3 SCFSI_{it} \times FR_{it} + \beta_4 BS_{ij} + \beta_5 LA_{it} + \beta_6 TCA_{it} + \beta_7 EA_{it} + \alpha_i + \delta_t + \varepsilon_{it}$$

(2)

i and t represent, banks and year indices; $a_i$ & $\delta_t$ represent firm & year level effect respectively; SCFSI is supply chain finance solutions index, FR is financial risk, BS is bank size, LA is leverage, TCA is tier 1 capital ratio, EA is earning assets, ai = country-level effect δt = year-level effect, β are the coefficients variables and ε = error term regression investigation in this study will focus on two main approaches: fixed and random effects models.

## Regression model-fixed-effect

The objective of this research, analyze the supply chain finance solutions impact on finance providers (banks) financial performance (H1) and moderating role of financial risk (H2). In the analysis, there were many challenges. First, although by adding many control variables like bank size, earning assets, tier-1 capital ratio, and loan-to-assets ratio, there are many other characteristics are unobservable which may relate the firm's decision to offer supply chain finance solutions and simultaneously its financial performance, raising possible concerns of endogeneity [49]. Second, this research sample time covered 10 years period of 2012 to 2021, during this time any event or trend like (COVID-10) is unobservable might also relation the financial performance. To address these challenges, First, all sample firms provided supply chain finance solutions between 2012–2021, with this firm-level fixed-effect estimation possible.

Furthermore, this research was interested in how firms' supply chain finance solutions affect their financial performance over time. Firm-level fixed effect approach used, there are many times variant firm characteristics that may link with the firm's supply chain finance solutions and financial performance of financial service providers such as firm culture and corporate decision-making; this will lead to reduced endogeneity concern and make within-firm consistent. Furthermore, there may be any event or trend effect which can effect model, to resolve this year-level fixed effect included. The fixed effect regression equation is:

$$ROA_{it} = \beta_o + \beta_1 SCFSI_{it} + \beta_2 FR_{it} + \beta_3 SCFSI_{it} \times FR_{it} + \beta_4 BS_{ij} + \beta_5 LA_{it} + \beta_6 TCA_{it} + \beta_7 EA_{it} + \alpha_i + \delta_t + \varepsilon_{it}$$

(3)

$$TOQ_{it} = \beta_o + \beta_1 SCFSI_{it} + \beta_2 FR_{it} + \beta_3 SCFSI_{it} \times FR_{it} + \beta_4 BS_{ij} + \beta_5 LA_{it} + \beta_6 TCA_{it} + \beta_7 EA_{it} + \alpha_i + \delta_t + \varepsilon_{it}$$

(4)

i and t represent, banks and year indices; $a_i$ & $\delta_t$ represent firm & year level effect respectively, and ε is an error term. *SCFSI* supply chain solutions index on financial service providers' financial performance (H1) and *FR* the financial risk moderating role (H2)

The first is the F-test to determine whether a mixed regression or fixed effect model should be used. The F-test p-value is 0.0000, which is significant and uses a fixed-effect model. Hausmann test p-value is 0.0000 rejecting the null hypothesis and using the fixed-effect model over the random-effect model.

## Results

Table 1 shows the correlation, mean and standard deviations of all variables. Fixed-effect model regression results are shown in Tables 2 and 3. There are 4 regression models in

**Table 1. Correlation matrix.**

|  | 1 | 2 | 3 | 4 | 5 | 6 | 7 | 8 |
|---|---|---|---|---|---|---|---|---|
| **ROA** | 1 | | | | | | | |
| **TOQ** | -0.2484 | 1 | | | | | | |
| **FR** | 0.2285 | -0.0044 | 1 | | | | | |
| **SCFSI** | -0.1616 | 0.1971 | -0.154 | 1 | | | | |
| **BS** | -0.2725 | -0.1241 | 0.146 | -0.1722 | 1 | | | |
| **EA** | 0.3205 | -0.1431 | -0.007 | 0.0670 | -0.6246 | 1 | | |
| **LA** | -0.4008 | -0.1259 | -0.157 | -0.1439 | 0.1275 | -0.336 | 1 | |
| **TCA** | 0.0499 | -0.0112 | 0.173 | -0.1495 | 0.1778 | -0.015 | -0.1232 | 1 |
| **Mean** | 0.0494 | -1.6861 | -1.1472 | 0.4171 | -0.3784 | 2.6197 | -0.8785 | -1.9238 |
| **Standard Deviation** | 0.0676 | 1.9846 | 0.6128 | 0.8506 | 0.7892 | 0.0715 | 0.3948 | 0.5329 |

Tables 2 and 3: Model-1 results show that control variables with bank and year level fixed effect; Model-2 results show that financial risk direct effect and Model-3 results show that direct effect of the supply chain finance solutions index. Model-4 results show that financial risk moderating role is statistically significant at ($p<0.01$), whereas F-statistics and R-squared 0.54 to 0.98 respectively.

Table 2 shows the results of the return of assets effect on supply chain finance solutions is significant ($p<0.10$); in model 4, its coefficient is 0.9888, and with Tobin's Q, its coefficient is 0.0106 means it has a positive relation with finance providers financial performance. This

**Table 2. Fixed effect test results of return on assets.**

| Variables | Model 1 | Model 2 | Model 3 | Model 4 |
|---|---|---|---|---|
| **Bank Size** | 1.2027 | 1.3994 | 1.6478 | 1.7313 |
| | (0.3913) | (0.3608) | (0.4282) | (0.4587) |
| **Loan to Asset** | -0.1303 | -0.1034** | -3.4643 | -3.8275 |
| | (0.3405) | (-0.8383) | (0.2504) | (0.2033) |
| **Tier 1 capita** | 0.90214** | 0.7466 | 0.8825* | 0.8387* |
| | (0.000) | (7.6918) | (0.0732) | (0.0946) |
| **Earning Assets** | 0.4638** | 0.1019 | 0.4192* | 0.4325 |
| | (0.000) | (0.000) | (0.0947) | (0.1414) |
| **Financial Risk** | | 0.4188** | 0.5694** | 0.0033 |
| | | (0.000) | (0.0288) | (0.9945) |
| **Supply Chain Finance Solution Index** | | | -0.0655 | 0.9888* |
| | | | (0.6372) | (0.078) |
| **Supply Chain Finance Solution Index × Financial Risk** | | | | 1.0209 (0.1092) |
| **Intercept** | -4.8821 | -5.2576 | -6.508 | -7.6466 |
| | 0.173 | 0.1782 | 0.3104 | 0.2203 |
| **Firm-level fixed effects** | Yes | Yes | Yes | Yes |
| **Year-level fixed effects** | Yes | Yes | Yes | Yes |
| **Number of observations** | 210 | 210 | 210 | 210 |
| **F-statistic** | 21.7109 | 23.4759 | 4.5317 | 4.5379 |
| **Prob(F-statistic)** | 0.000 | 0.000 | 0.000 | 0.000 |
| **Adjusted R$^2$** | 0.7659 | 0.7852 | 0.5446 | 0.5566 |

Note: 10% (*) and

5% (**)

**Table 3. Fixed effect test results of Tobin's Q.**

| Variables | Model 1 | Model 2 | Model 3 | Model 4 |
|---|---|---|---|---|
| **Bank Size** | 0.0034 | 0.0037 | 0.0127 | 0.0133 |
| | (0.6844) | (0.6472) | (0.2851) | (0.2064) |
| **Loan to Asset** | -0.0002 | -0.0002 | -3.4643 | -0.0337 |
| | (0.7806) | (0.8291) | (0.2504) | (0.3468) |
| **Tier 1 capita** | 0.0003 | -4.2507 | -0.0037 | -0.0040 |
| | (0.8044) | (0.9997) | (0.3672) | (0.3486) |
| **Earning Assets** | 0.0043** | 0.0044** | 0.0032 | 0.0032 |
| | (0.0000) | (0.0000) | (0.3665) | (0.3559) |
| **Financial Risk** | | 0.0007* | 0.0011* | -0.0026** |
| | | (0.0656) | (0.007) | (0.026) |
| **Supply Chain Finance Solution Index** | | | 0.0037 | 0.0106** |
| | | | (0.1401) | (0.0000) |
| **Supply Chain Finance Solution Index × Financial Risk** | | | | -0.0067*** (0.0029) |
| **Intercept** | 0.0118 | 0.0111 | -0.0175 | -0.0250 |
| | (0.571) | (0.5857) | (0.7843) | (0.6924) |
| **Firm-level fixed effects** | Yes | Yes | Yes | Yes |
| **Year-level fixed effects** | Yes | Yes | Yes | Yes |
| **Number of observations** | 210 | 210 | 210 | 210 |
| **F-statistic** | 318.3026 | 309.526 | 164.7493 | 161.2075 |
| **Prob(F-statistic)** | 0.0000 | 0.0000 | 0.0000 | 0.0000 |
| **Adjusted R$^2$** | 0.9804 | 0.9805 | 0.9823 | 0.9827 |

Note: 10% (*)

5% (**) and

1% (***)

means that a 1% increase in supply chain finance solutions will also increase financial performance by 0.9888 and 0.0106, respectively. Thus, H1 is accepted based on model number 4.

Financial risk as a moderating effect, the interaction between supply chain finance solutions and financial risk is significant (p<0.01), as shown in model 4 Table 3. The coefficient is negative 0.0067, which means the moderating effect is interference rather than reinforcement. Financial risk reduces when finance providers offer SCF solutions with reinforcement of financial performance. In Table 2, the coefficient is positive 1.0209. Financial risk as moderating effect reinforcement not an interference.

## Conclusion

This research reveals the supply chain fiancé solutions effect on finance provide financial performance. First, it demonstrates that finance provider finance performance may enhance by offering supply chain finance supply chain finance solutions enhance the financial performance of financial service providers. Moreover, financial service providers' financial performance financial risk mitigates with supply chain finance solutions. As per [10], finance provider financial performance increase and relaxation in financing term is capital level strategy of supply chain finance. Following the literature pattern, with the help of supply chain finance solutions, financial service providers' financial risk can mitigate and enhance financial performance because financial risk impacts financial service providers and overall stakeholders [50]. It has to consider for supply chain finance solutions development and implementation.

Therefore, the research findings indicate that financial service providers can use supply chain finance solutions as a strategic move to enhance financial performance and mitigate their financial risk, as shown in Tables 2 and 3 (model 4).

## Implications of research

Future research and implication of this research, first, some researchers have emphasized that supply chain finance solutions can provide a "win-win" situation, the financial risk implication of SCF solutions and its impact of finance providers financial performance are not clear. This is the unique study, how empirically supply chain finance solution index affects financial service providers' financial performance with financial risk. Scholars in finance allow them to extend the finance literature boundaries and investigate supply chain finance solutions associations with non-financial strategies linked with financial risk.

## Supporting information

**S1 Annex.**
(DOCX)

## Author Contributions

**Conceptualization:** Munibah Munir.

**Data curation:** Munibah Munir.

**Formal analysis:** Munibah Munir, Nousheen Tariq Bhutta.

**Investigation:** Munibah Munir.

**Methodology:** Munibah Munir, Nousheen Tariq Bhutta.

**Project administration:** Munibah Munir.

**Resources:** Munibah Munir.

**Software:** Munibah Munir.

**Supervision:** Nousheen Tariq Bhutta.

**Writing – original draft:** Munibah Munir.

**Writing – review & editing:** Nousheen Tariq Bhutta.

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
