## [Decision Letter · Decision Letter 0]

27 Jul 2023

PONE-D-23-15522Light in the Tunnel or Just a train; Impact of Supply Chain Finance Solutions on Financial Service Providers' Financial Performance by Mitigating Financial Risk.PLOS ONE

Dear Dr. Munir,

Thank you for submitting your manuscript to PLOS ONE. After careful consideration, we feel that it has merit but does not fully meet PLOS ONE’s publication criteria as it currently stands. Therefore, we invite you to submit a revised version of the manuscript that addresses the points raised during the review process.

We look forward to receiving your revised manuscript.

Kind regards,

João Zambujal-Oliveira

Academic Editor

PLOS ONE

Journal Requirements:

https://livrepository.liverpool.ac.uk/3118010/1/IJOPM-07-2020-0462.R3.pdf

https://www.cell.com/heliyon/fulltext/S2405-8440(22)00248-1?_returnURL=https%3A%2F%2Flinkinghub.elsevier.com%2Fretrieve%2Fpii%2FS2405844022002481%3Fshowall%3Dtrue

In your revision ensure you cite all your sources (including your own works), and quote or rephrase any duplicated text outside the methods section. Further consideration is dependent on these concerns being addressed.

"The Funders had no role in study design, data collection and analysis, decision to publish or preparation of the manuscript. "

"I have read the journal's policy and authors of this manuscript have no competing interests."

**Additional Editor Comments:**

Please take into consideration all the comments provided by the two reviewers and update the paper to address the identified issues.

After making the necessary revisions, the paper should be resubmitted for verification to ensure that it meets the conditions required by the reviewers for acceptance.

Reviewers' comments:

Reviewer's Responses to Questions

**Comments to the Author**

1. Is the manuscript technically sound, and do the data support the conclusions?

Reviewer #1: Partly

Reviewer #2: Yes

2. Has the statistical analysis been performed appropriately and rigorously? 

Reviewer #1: No

Reviewer #2: Yes

3. Have the authors made all data underlying the findings in their manuscript fully available?

Reviewer #1: No

Reviewer #2: Yes

4. Is the manuscript presented in an intelligible fashion and written in standard English?

Reviewer #1: Yes

Reviewer #2: No

5. Review Comments to the Author

Reviewer #1: Comments on the paper “Light in the Tunnel or Just a train; Impact of Supply Chain Finance Solutions on Financial Service Providers' Financial Performance by Mitigating Financial Risk”

After reading the paper, I found that the author had a good research idea and research questions to explore. However, I found there are some issues with the execution of the research. I am listing my comments below:

1. The author used dummy variables to denote the Supply Chain Finance (SCF) solutions offerings by Financial Service Providers (FSPs). If the FSP offer a particular SCF solution in a given year, then it is coded as 1, else 0. Thus, reducing the SCF solutions offering to binary variables. However, I have a main concern of using Principal component Analysis (PCA) for data reduction or the creation of SCF index. PCA is suitable for reducing the numeric multivariate variables. If it is a mixed of numeric variables and binary variables, PCA can still be useful. There is no justification on the appropriateness of the PCA for reducing binary variables. Further, the paper makes no comment on the number of SCF solutions offered by banks. What was the maximum number of SCF solutions offered by FSPs? It is not clear out of how many binary variables (SCF solutions), the SCF Index was created.

2. “Using principal component analysis (PCA), the Eviews index is created”- The writing needs to be clearer. What is Eviews index? Do you mean SCF solution index is generated using PCA on Eviews software?

3. “For example, if a bank offers account receivables finance or an early payment discount program, it is coded as 1 and 0 if not offered buyer direct financing”. I have a concern with “0 if not buyer direct financing”. What does the author intend by this statement? Does the author mean that if the FSP is not offering any of the SCF solutions, then offering of buyer direct financing is assumed? If so then it could be wrong assumption. Just because a particular firm does not avail SCF solution from the FSP, does not mean it will always get buyer direct financing. A firm getting buyer direct financing may also simultaneously avail SCF solution from the bank.

4. Table 2a: The Adjusted R2 keeps on declining when new variables are added (Models 3 and 4). It means the main variables SCF solution index and SCF x Financial Risk are not contributing to the overall model. How do you reconcile with the overall results?

Table 2b and its interpretation: “The coefficient is negative 0.0067, which means the moderating effect is interference rather than reinforcement. This suggests that the bank’s supply chain finance solution increases financial performance by increasing financial risk and supporting H2, which is the Supply chain finance solution that enables the financial performance of financial service providers to mitigate their financial risk”. The author states that the coefficient is negative 0.0067 whereas Table 2b shows it to be positive. The interpretation “This suggests that the bank’s supply chain finance solution increases financial performance by increasing financial risk and supporting H2, which is the Supply chain finance solution that enables the financial performance of financial service provider to mitigate their financial risk” is questionable.

5. More care needs to be given in writing the overall manuscript.

Given the above issues, I feel the manuscript is not fit to be accepted at its current state. A lot of reworking will be required.

Reviewer #2: English editing needed to make the document more readable. There are some paragraphs hard to read.

There are also some parts that need more description and explanation as the items presented have not been fully documented and introduced previously.

6. PLOS authors have the option to publish the peer review history of their article (what does this mean?). If published, this will include your full peer review and any attached files.

Reviewer #1: No

Reviewer #2: **Yes: **Enrico Camerinelli

---

## [Author Response · Author response to Decision Letter 0]

30 Aug 2023

Comment to Editor:

1 Please ensure that your manuscript meets PLOS ONE's style requirements, including those for file naming. 

Done.

2. We noticed you have some minor occurrence of overlapping text with the following previous publication(s), which needs to be addressed

Done

5. In your Data Availability statement, you have not specified where the minimal data set underlying the results described in your manuscript can be found. "

Data Availability Sources provided.

Reviewer 1:

First, thank you for your time of reviewing the research paper with valuable suggestions. All the suggestions and comments has been incorporated, which improve the document. The revised paper is being submitted with a request to consider this for publication. 

Comments to the Author: The author used dummy variables to denote the Supply Chain Finance (SCF) solutions offerings by Financial Service Providers (FSPs). If the FSP offer a particular SCF solution in a given year, then it is coded as 1, else 0. Thus, reducing the SCF solutions offering to binary variables. However, I have a main concern of using Principal component Analysis (PCA) for data reduction or the creation of SCF index. PCA is suitable for reducing the numeric multivariate variables. If it is a mixed of numeric variables and binary variables, PCA can still be useful. There is no justification on the appropriateness of the PCA for reducing binary variables. Further, the paper makes no comment on the number of SCF solutions offered by banks. What was the maximum number of SCF solutions offered by FSPs? It is not clear out of how many binary variables (SCF solutions), the SCF Index was created.

Author Response: Change has been incorporated into the document. The Principal Component Analysis technique of (Baker & Wurgler, 2007), is applied to quantify the supply chain finance solutions index using supply chain finance solutions as proxies. There is maximum 21 number of SCF solutions offered by FSPs. 

Comment to the Author: “Using principal component analysis (PCA), the EViews index is created”- The writing needs to be clearer. What is views index? Do you mean SCF solution index is generated using PCA on views software?

Author Response: Change has been incorporated into the document. Supply chain finance solution index which is generated by using PCA on EViews software is correct statement. 

Comment to the Author: For example, if a bank offers account receivables finance or an early payment discount program, it is coded as 1 and 0 if not offered buyer direct financing”. I have a concern with “0 if not buyer direct financing”. What does the author intend by this statement? Does the author mean that if the FSP is not offering any of the SCF solutions, then offering of buyer direct financing is assumed? If so then it could be wrong assumption. Just because a particular firm does not avail SCF solution from the FSP, does not mean it will always get buyer direct financing. A firm getting buyer direct financing may also simultaneously avail SCF solution from the bank.

Author Response: Change has been incorporated into the document. Each bank website check and mark how many supply chain finance solutions offered by particular bank out of list of the supply chain finance solutions, if a bank offers any one solution like account receivables finance or an early payment discount program it is coded as 1. On the other side if bank is not offering any particular supply chain finance solution it is coded as 0.

Comment to the Author: Table 2a: The Adjusted R2 keeps on declining when new variables are added (Models 3 and 4). It means the main variables SCF solution index and SCF x Financial Risk are not contributing to the overall model. How do you reconcile with the overall results?

Author Response: Table 2a financial performance measure with the proxy return on assets, the Adjusted R2 in table 2a keeps on declining the adjusted r-squared response for new variable and increase if new variable enhances the model above about what would be obtained by probability. On the other side it is decline as SCF x Financial Risk improves the model less than what is predicted. When new variable added in the model, adjusted R2 will automatically decrease. 

Comment to the Author: “The coefficient is negative 0.0067, which means the moderating effect is interference rather than reinforcement. This suggests that the bank’s supply chain finance solution increases financial performance by increasing financial risk and supporting H2, which is the Supply chain finance solution that enables the financial performance of financial service providers to mitigate their financial risk”. The author states that the coefficient is negative 0.0067 whereas Table 2b shows it to be positive. The interpretation “This suggests that the bank’s supply chain finance solution increases financial performance by increasing financial risk and supporting H2, which is the Supply chain finance solution that enables the financial performance of financial service provider to mitigate their financial risk” is questionable.

Author Response: Change has been incorporated into the document. The coefficient is negative 0.0067, which means the moderating effect is interference rather than reinforcement. Financial risk reduces when financial service providers offer supply chain finance solutions with reinforcement of financial performance.

Comment to the Author: More care needs to be given in writing the overall manuscript.

Author Response: Change has been incorporated into the document. Prof-read all the document and made necessary changes in the manuscript. 

Reviewer 2

First, thank you for your time of reviewing the research paper with valuable suggestions. All the suggestions and comments has been incorporated, which improve the document. The revised paper is being submitted with a request to consider this for publication. 

Comment to the Author: English editing needed to make the document more readable. There are some paragraphs hard to read.

Author Response: Change has been incorporated into the document. Rewrite the paragraphs to make it more clear reading.

---

## [Decision Letter · Decision Letter 1]

21 Sep 2023

Light in the Tunnel or Just a train; Impact of Supply Chain Finance Solutions on Financial Service Providers' Financial Performance by Mitigating Financial Risk.

PONE-D-23-15522R1

Dear Dr. Munir,

We’re pleased to inform you that your manuscript has been judged scientifically suitable for publication and will be formally accepted for publication once it meets all outstanding technical requirements.

Kind regards,

João Zambujal-Oliveira

Academic Editor

PLOS ONE

Additional Editor Comments (optional):

Reviewers' comments:

Reviewer's Responses to Questions

**Comments to the Author**

1. If the authors have adequately addressed your comments raised in a previous round of review and you feel that this manuscript is now acceptable for publication, you may indicate that here to bypass the “Comments to the Author” section, enter your conflict of interest statement in the “Confidential to Editor” section, and submit your "Accept" recommendation.

Reviewer #2: All comments have been addressed

2. Is the manuscript technically sound, and do the data support the conclusions?

Reviewer #2: Yes

3. Has the statistical analysis been performed appropriately and rigorously? 

Reviewer #2: N/A

4. Have the authors made all data underlying the findings in their manuscript fully available?

Reviewer #2: Yes

5. Is the manuscript presented in an intelligible fashion and written in standard English?

Reviewer #2: Yes

6. Review Comments to the Author

Reviewer #2: (No Response)

7. PLOS authors have the option to publish the peer review history of their article (what does this mean?). If published, this will include your full peer review and any attached files.

Reviewer #2: **Yes: **Enrico Camerinelli

---

## [Editor Report · Acceptance letter]

26 Sep 2023

PONE-D-23-15522R1 

Light in the Tunnel or Just a train; Impact of Supply Chain Finance Solutions on Financial Service Providers' Financial Performance by Mitigating Financial Risk. 

Dear Dr. Munir:

I'm pleased to inform you that your manuscript has been deemed suitable for publication in PLOS ONE. Congratulations! Your manuscript is now with our production department. 

Kind regards, 

on behalf of

Prof. João Zambujal-Oliveira 

Academic Editor

PLOS ONE